# ConvBench: A Comprehensive Benchmark for 2D Convolution Primitive Evaluation

Lucas Alvarenga, Victor Ferrari, Rafael Souza, Marcio Pereira, Guido Araujo

Instituto de Computação, Universidade Estadual de Campinas - (UNICAMP), Campinas, SP – Brazil.

*lucas.silva@ic.unicamp.br, v187890@dac.unicamp.br, rafael.souza@ic.unicamp.br,*
*mpereira@ic.unicamp.br, guido@unicamp.br*

*Abstract*—Convolution is a compute-intensive operation placed at the heart of Convolution Neural Networks (CNNs). It has led to the development of many high-performance algorithms, such as Im2col-GEMM, Winograd, and Direct-Convolution. However, the comparison of different convolution algorithms is an error-prone task as it requires specific data layouts and system resources. Failure to address these requirements might lead to unwanted time penalties. Thus, considering all processing steps within convolution algorithms is essential to comprehensively evaluate and fairly compare their performance. Furthermore, most known convolution benchmarking adopts ad-hoc testing suites with limited coverage and handmade operations. This paper proposes ConvBench, a primitive-level benchmark for the evaluation and comparison of convolution algorithms. It assesses 9243 convolution operations derived from 1097 real-world deep learning models, resulting in performance and execution breakdown graphs for a detailed evaluation. ConvBench capability is evaluated across the Sliced Convolution (SConv) algorithm. The experiments showed results faster than Im2col-GEMM in 93.6% of the convolutions. However, the use of ConvBench allowed the delving into the remaining 6.4% underperforming convolutions, uncovering a critical slowdown of 79.5% on average of SConv's packing step. This analysis underscores a potential source of optimization for SConv, opening up new paths for convolution designers to improve their algorithms.

## I. Introduction

The convolution operation is a compute- and memory-intensive operation that may account for up to $90\%$ of a Convolutional Neural Network (CNN) runtime [9]. It strongly motivated the search and development of various high-performance algorithms for the convolution operator, which can be roughly divided into Fast Fourier Transform and Winograd [17], GEneral Matrix Multiply (GEMM) [6], [15], and Direct Convolution (DC) [4], [5], [10], [12], [19] approaches.

However, state-of-the-art convolution algorithms had assessed their approach in a minimal and narrow set of convolution operations, lacking completeness and possibly avoiding commonly found convolutions, such as strided/padded operations. For example, some works only use a small and fixed set of handmade convolutions [4], [16]. Others, on the other hand, leverage more realistic conditions as they assemble their operation set from off-the-shelf Deep Learning (DL) models, such as from the ResNet-50, AlexNet, VGG16, Inception-V3 and GoogleNet [5], [6], [10], [12], [15], [19]. Yet, their testing operation set account to a small set of at most 134 convolution operations.

The algorithms are also not directly comparable. Different algorithms have their own premises of data layout and system resources that complicate the fair comparison of timing measurements. Simpler Im2col-GEMM techniques require a unique pre-convolution reordering routine, which adds a penalty to the final operation time; DC-based approaches dilute the GEMM-required packing routine in the convolution execution time, repeating the routine execution and possibly inducing calling overheads [6], [10]; Approaches that leverage Just-in-Time [14] compilation techniques may incur pre-convolution analysis and compilation time penalties that must be diluted through operation reuse. These penalties worsen the final operating time, leading to unfair comparison results. Hence, to compare different algorithms, it is important to analyze their execution in its entirety, discriminating and comparing all convolution-related processing steps.

This article proposes ConvBench, a convolution benchmark that tackles both the completeness and fairness problems. It addresses the **completeness** problem by evaluating the algorithm in a Convolution Operation Set (*convSet*) containing 9243 unique 2D convolution operations from 1097 real DL models. The **fairness** problem is handled by the use of a Timing Measurement Tool (TM-Tool) aligned with a standardized timing nomenclature, which identifies the encountered steps of a convolution algorithm (*e.g.* packing, tiling, etc.). Finally, ConvBench generates insightful execution breakdown and performance graphs of the evaluated convolution algorithm.

Three main steps must be performed in order to use ConvBench: (1) the construction of the convSet; (2) the convolution algorithm integration and timing measurement tool adoption, and (3) the execution and final result summarization with graph generation. In this work, ConvBench usage is evaluated across the Sliced Convolution (SConv) [10] algorithm. At first, the ConvBench's convSet coverage is evaluated by juxtaposing it against the testing range of SConv. Particularly, the SConv test suite contains 134 operations, representing 1.5% of ConvBench's convSet. In sequence, ConvBench usage steps (2) and (3) are evaluated by integrating the SConv experimental protocols within ConvBench. This evaluation uncovered that SConv's packing step is on average 79.5% slower when Im2col-GEMM outperforms SConv. This finding exposed a performance issue, which may aid convolution designers in improving their overall algorithm's performance.

The remainder of this paper presents, respectively, a brief Related Works, the ConvBench Internals, SConv results within ConvBench, and, finally, Conclusions and Future Directions.

## II. RELATED WORK

This section presents an overview of DL benchmarks from the literature, describing their scope and characteristics.

DL benchmarks can be roughly divided based on the granularity of their measurement scope, such as model-wise, layer-wise, and primitive-scoped granularity. Model-wise benchmarks [8], [18] keep the underlying task unchanged while time-to-convergence, latency, and throughput metrics are assessed for the DL model. Layer-wise benchmarks [7], [13] assess the performance of DL models based on execution time information and memory signature of both the entire model (*i.e.* model-wise granularity) and each of the model's building blocks (*i.e.* layer-wise granularity). Primitive-scoped benchmarks [1], [11], [14], on the other hand, have been designed to be aware of the underlying system, bringing internal system aspects to the analysis at the cost of being Neural-Network library (NN-lib) dependent. Finally, all acquired information is commonly summarized in tables and visualization charts to draw attention to optimization opportunities.

A naive approach to evaluate convolution operations would rely on primitive-scoped benchmarks to assess their performance. However, current benchmarks neither provide a specialized convolution metric measurement (fairness problem) [1], [14] nor assess its algorithms in a wide operation set (completeness problem) [11].

Considering the aforementioned issues, this article proposes ConvBench, a benchmark for evaluating and comparing convolution primitives from different sources (*e.g.* NN-lib, frameworks, etc.) in a comprehensive set of real convolution operations with a specialized set of measurements.

## III. CONVOLUTION BENCHMARKING TOOL: CONVBENCH

This section describes the proposed Convolution Benchmarking Tool, ConvBench. It leverages the TM-Tool as a central entity and requires the execution of three main steps: (a) the convSet construction; (b) the convolution algorithm integration, and (c) the result visualization.

### A. The Timing Measurement Tool – TM-Tool

In general, it is possible to identify common structures across different convolution algorithms. As shown in Fig. 1, this work divides the convolution operation into three main moments: the pre-, in-, and post-convolution processing steps. For each of the processing steps, some common routines could be identified. The **pre-convolution** step comprises the operation analysis and data reordering routines. **in-convolution** covers routines related to the convolution execution, such as the data tiling, the microkernel-specific packing, the microkernel execution, and result unpacking. Finally, the **post-convolution** processing step possibly executes an additional reordering routine over the results from the in-convolution step.

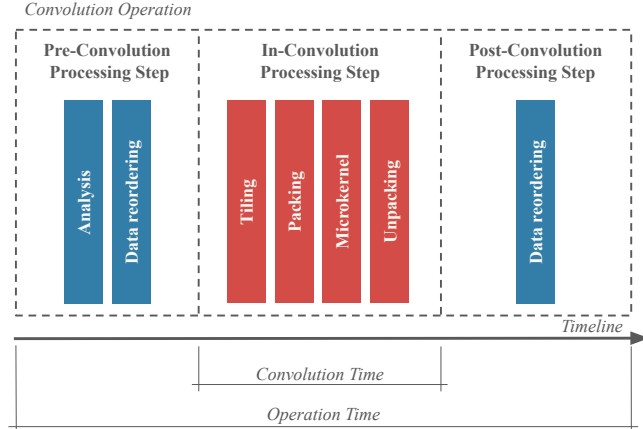

Fig. 1. Proposed Time Measurement Tool. The name of the class start/update API methods follows the same processing step names.

Hence, to provide fair convolution algorithm comparisons, a Timing Measurement tool has been developed as a central entity of ConvBench. The TM-Tool is a class based on the `chrono` C++ library with an easy-to-use Application Programming Interface (API) that can be leveraged in different convolution implementations. TM-Tool assumes the aforementioned routine names for its start/update time-measure methods. Each method alters internal states that hold the elapsed time and number of calls measurement information. Additionally, it also contains an **Operation Time** and a **Convolution Time** measure, which refers to the summation of all three processing steps and the summation of all in-convolution routines, respectively.

To leverage the TM-Tool capability in ConvBench, the operation processing steps must be enclosed with the related TM-Tool method. The API call displacement must be carefully added since the resulting values are central to the ConvBench final analysis. See Listing 1 for a concrete example.

### B. Convolution Operation Set Construction and Filtering

This step describes the first main task of ConvBench: the acquisition, construction, and filtering of the convSet.

The convSet is constructed by iterating over each available model from Hugging Face TIMM and PyTorch Torchvision DL model collections [2], [3]. As of now, TIMM's model collection has 1017 DL models and PyTorch contains 80 DL models, comprised of different kinds of CNNs and Visual Transformers, such as ResNet, DarkNet, EfficientNet, MaxViT, FastViT. For each of the 1097 models: (1) A randomly valued input that meets the model's input shape criteria is generated. Then, (2) all model's 2D convolution layers are encapsulated by a structure that captures the layer parameters and the input and output shape. Finally, (3) the generated input is propagated through the adjusted model and shape information is extracted. At this point, an identifying key is created based on the layer shape information. This key is then added to the convSet only if there is no equal key, avoiding operation duplication.

The final convSet contains a total of 9243 different convolution operations, among which:

- 5622 are pointwise ($1 \times 1$ filter) convolution operations.
- Of the remaining 3620 convolutions:
  - 2366 are grouped convolutions;
  - 17 are dilated convolutions;
  - 95 have rectangular filters;
  - 1213 are related to **regular** convolution operations (*i.e.* not pointwise, dilated, grouped, and rectangular filtered operations).

The convSet contains operations with input, output, and filter spatial shapes of up to $1024 \times 1024$, $400 \times 400$, and $32 \times 32$, respectively. Input and output channels vary from 1 to 12288. The diverse span of convSet presents computationally cheap operations with a few hundred Float Point Operations (FLOPS) up to expensive operations with more than $10^9$ FLOPS. It enables convolution algorithm assessment across a wide range of operations, aiding convolution designers to identify and tackle optimization hotspots.

However, optimized convolution implementations usually do not accept all possible operation configurations. SConv, for example, does not address grouped and dilated operations. For this reason, ConvBench provides a convSet filtering mechanism, easily selecting only the desired convolution operations. The filtered convSet is then exported to a Comma Separated Value (CSV) file that will be read during the execution. For more information, see the ConvBench GitHub repository.[1]

### C. ConvBench Usage

The ConvBench is designed as a C++ abstract and pure virtual class in the form of a header file that must be included and inherited by the user-defined subclass. The description of the ConvBench main methods are:

- `virtual void convolution(...)`: The main convolution algorithm to be assessed. This is an abstract method and must be overridden by the subclass.
- `virtual void convolution_baseline(...)`: The baseline convolution algorithm. This method is set to an Im2col-GEMM implementation by default, but can be overridden by its subclass.
- `void convset_exec(...)`: The execution procedure method. When called, this method iterates through the filtered convSet, creating the required convolution data and executing the chosen convolution implementation.

The outline of a user-defined ConvBench's subclass in C++ can be seen on Listing 1. At first, it includes `convbench.h`. Then a novel class is derived by inheriting from `ConvBench` class with the desired `T` data type. The selection of `T` controls the data type and, consequentially, the size of the input, output, filters, and bias buffers. In sequence, `convolution` and `baseline_convolution` methods are overridden by the subclass, containing the desired convolution implementation.

[1]https://github.com/LucasFernando-aes/ConvBench

An example of how to use the TM-Tool API is shown in the `convolution` method. The start/update TM-Tool API method encloses a possible pre-convolution data reordering routine. In the end, a simple `main` method is provided. It instantiates the benchmark subclass object and starts the experimental procedure with a `convset_exec` method call.

The `convset_exec(...)` method expects two arguments: a *data generation strategy* option and an *execution* option. The data generation strategy defines how the data buffers will be filled, *i.e.*, with random (`random` option) or constant (`constant` option) values. The execution option defines which convolution implementation will be evaluated. `main` option executes the user-defined `convolution(...)` method, `baseline` option executes the `convolution_baseline(...)` method, and `correctness` option evaluates the correctness of the main execution option in regard the baseline output. ConvBench also accepts experimental configuration options such as the number of `warm-ups` executions that precede the number of measured executions (`runs` option). Otherwise stated, all experiments used the `random` data generation strategy.

Speedup analysis occurs in a two-way step outside the ConvBench executing scope. A script separately executes the ConvBench with the `main` option and the `baseline` option. Moreover, the actual ConvBench version only measures the execution time of the convolution-specific aforementioned routines. The next versions of ConvBench envisage the measurement of system-related metrics, such as cache hit/misses, Translation Lookaside Buffer misses, and memory accesses.

```cpp
#include "convbench.h"

template <T>
class Inherited_Conv : public ConvBench<T>{
    // Constructors and destructors
    Inherited_Conv() : ConvBench<T>() {...};
    ~Inherited_Conv();

    // Main convolution algorithm implementation
    void convolution(args...) override {
        //example of TM-Tool usage.
        TIME(preconv_packing_start())
        input_packing(args...);
        TIME(preconv_packing_update())

        // Possible other statements and expressions
        ...
    }

    // Baseline convolution algorithm implementation
    void convolution_baseline(args...) override {
    ... }

    // Possible other methods ...
}
int main(...) {
    // Object instantiation
    Inherited_Conv<T> bench = Inherited_Conv<T>();
    // Benchmark execution
    bench.convset_exec(args...);
}
```

Listing 1. ConvBench Usage Example

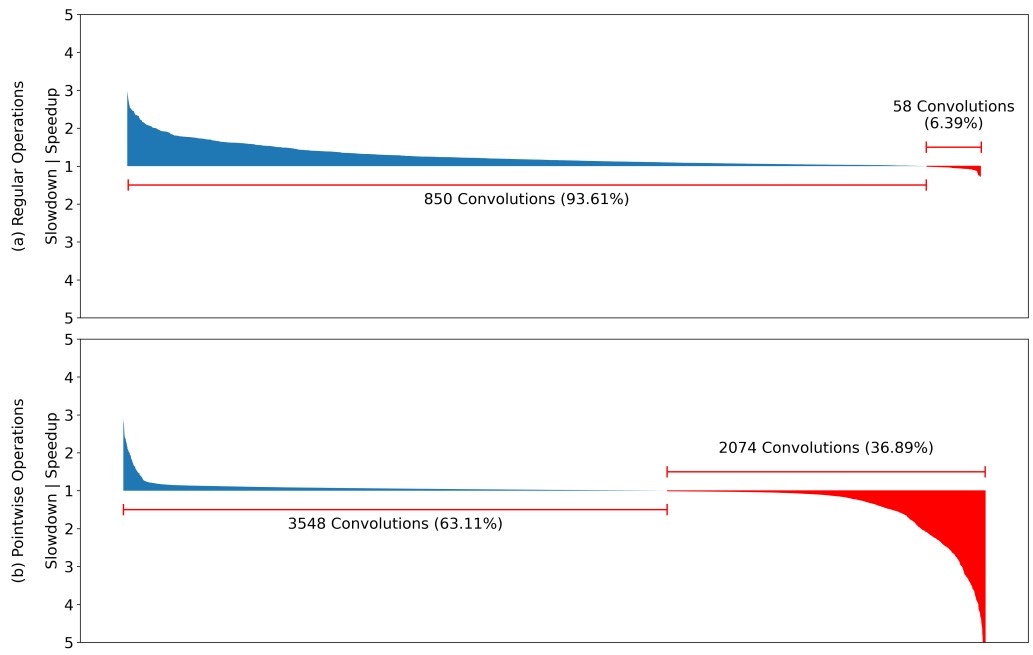

Fig. 2. In-Convolution speedup/slowdown analysis of Sconv against Im2col-GEMM. (a) refers to the filtered convSet only containing regular operations without zero-padded operations, while (b) refers to the pointwise filtered convSet.

### D. Result Acquisition and Visualization

At the end of the execution, ConvBench produces a new CSV file containing the measured timings and number of API calls. With this file, some plots are automatically generated:

- A speedup/slowdown barplot to detail the speedup magnitude of the main algorithm regarding the baseline.
- A breakdown graph of execution time which compares the time spent on the main and baseline implementations.

The former plot highlights which cases and by how much the main convolution implementation surpasses the baseline measurements. The latter emphasizes implementation optimization opportunities by showing the time breakdown of each processing step of the convolution algorithm. Examples of final charts can be seen in the next Section IV.

## IV. EXPERIMENTS AND RESULTS

This section showcases the evaluation of SConv [10] within ConvBench. It first compares the experimental range of SConv against ConvBench and, then, extends its experimental procedure to the largest possible filtered subset of convSet, *i.e.*, the pointwise and regular filtered versions of convSet.

SConv [10] is a recent DC algorithm based on the MLIR/L-LVM code-generation toolchain. It consists of a system-agnostic compile-time Convolution Slicing Analysis (CSA) step, followed by architecture-specific macro-kernel code generation Convolution Slicing Optimization (CSO) and input-tensor packing Vector Based Packing (VBP) steps. The authors used the Im2col-GEMM convolution algorithm as the baseline implementation and evaluated their approach in 134 convolution operations acquired from 7 common real-world DL models.

*a) Experimental procedure:* All experiments were evaluated in a computer node containing an Intel Xeon Silver 4208 processor fixed in performance mode at 2.10 GHz. It contains 8 cores with Hyper-Threading enabled, Avx512 vector extension, an L1 cache of 32kB, L2 cache of 1MB, and L3 cache of 11MB. The system's operations system is a Rocky Linux 8.8 with kernel 4.18.0. SConv experiments were compiled by a Clang 14 compiler with *O3*, *ftree-vectorize*, and *finline-functions* options. Moreover, the same protocol experimental of SConv is adopted: each experiment is limited to a single-thread execution of individual input instances (*i.e.* batch size of 1). Next, all experiments were repeated 100 times with warm-up rounds before the effective operation assessment.

*b) Operation juxtaposing analysis:* SConv original results were assessed on 134 unique convolutions acquired from 7 real DL models, the GoogleNet, Inception-V2, ResNet-18/50/152, SqueezeNet and VGG16. 72 operations out of the 134 refer to pointwise convolutions and the other 62 refer to regular convolutions. Particularly, 114 of them match operations from ConvBench, while the remaining 20 operations refer to convolutions from the former GoogleNet and Inception-V2 DL models. ConvBench contains the most recent version of these models (*e.g.* Inception-V3).

*c) Results:* Fig. 2 shows SConv speedup/slowdown results regarding the baseline in ConvBench's regular and pointwise convSet. SConv outperforms the baseline in 93.6% of ConvBench's regular operations and 63.1% for the pointwise convSet. The former results match the authors' findings, showing a great generability level. On the other hand, the latter results showed a minor ratio of speedup cases, which is expected since SConv was not optimized for such operations.

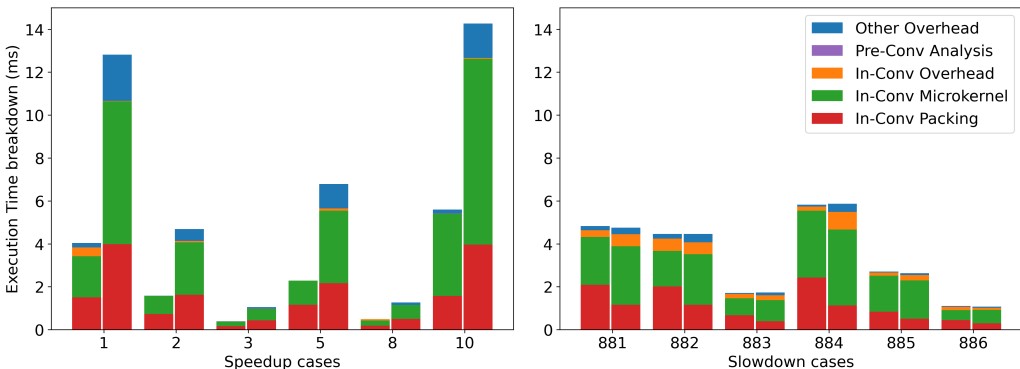

Fig. 3. Execution breakdown graph. The X-axis indicates the operation index within the convSet, while the Y-axis shows the time magnitude in milliseconds. Particularly, For a given index, Left-sided bars refer to SConv measured timings, and right-sided bars refer to Im2col+GEMM baseline timings.

Additionally, some insights can be drawn from the time breakdown graph in Fig 3. For example, there are $474$ cases in which SConv's packing step underperforms the packing step of Im2col-GEMM. The SConv packing step maintains an average speedup of $53.77\%$ against Im2col packing step whenever it outperforms the baseline. Conversely, when SConv underperforms the baseline, an average slowdown of $79.5\%$ could be observed. Focusing the analysis on SConv's packaging routine exposes that underperforming cases only occur in operations with stride different from 1 and a small number of channels. These convolutions do not leverage the author's VBP strategy, resorting to a non-optimized and scalar packing routine. This information aids the author in identifying and focusing on the main issue, potentially addressing the performance bottleneck.

## V. CONCLUSION

Convolution is an operation placed at the heart of CNNs that demands significant computational resources. However, the comparison of different convolution algorithms imposes some challenges. This work proposes ConvBench, a specialized benchmark for evaluating and comparing convolution algorithms across 9243 convolution operations acquired from real-world CNN models. The applicability of Convbench on SConv has shown that the In-Convolution packing routine is a bottleneck when SConv underperforms the baseline. This observation became possible given the comprehensive evaluation and detailed comparisons of ConvBench, which help convolution designers in identifying optimization opportunities. Future steps of ConvBench expect to expand the TM-Tool in the direction of a profiling encapsulation API, providing the assessment of both timing and system metrics; and evaluate the ConvBench in other hardware systems, particularly, RISC-V and PowerPC architectures.

## ACKNOWLEDGEMENTS

This research was supported by São Paulo Research Foundation - FAPESP (grants 2023/03328-9) and UNICAMP.

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
