# OpenReview forum: "ConvBench: A Comprehensive Benchmark for 2D Convolution Primitive Evaluation"
_iscaconf.org/ISCA/2024/Workshop/MLArchSys — MLArchSys 2024 OralPoster_

### Official Review · Reviewer_tgGx · 2024-05-27
**ConvBench: A Comprehensive Benchmark for 2D Convolution Primitive Evaluation**

**Confidence:** 3
**Rating:** 6

**Detailed Feedback And Questions For Authors:**

* The introduction argues that ConvBench is novel due to its ability to address the completeness problem and the fairness problem. This feels a bit disconnected from the presentation of benchmarks in the related works section, where the approach of previous benchmarks to these problems could be described in greater detail but neither of these two problems are mentioned again.
* Some minor grammar/formatting inconsistencies (e.g. "It"/"it" in intro 3rd paragraph, missing "and" in Section 3A last paragraph, etc.). The use of "displaced" seems strange in the conclusion.
* The end of the introduction states that the results have helped convolution designers improve SConv, while the results section puts this only as a suggestion "...developers *may* further investigate..." Consistency and clarity here would be good.
* Is there any meaning that should be drawn from the values in the x-axis of Fig 3?
* The introduction mentions a Future Work section, which does not seem to be present. There are some words in the conclusion referring to future work, but it would be helpful to expand on this some more.
* (It would be great to have the code for the benchmark released in the future.)

**Top Reasons To Accept The Paper:**

* This paper presents a new benchmark for evaluating convolution algorithms. The benchmark has high coverage of convolution algorithms and is able to uncover optimization opportunities in SConv.
* The authors formulate the motivations well by describing the completeness and fairness problems. The methods, experiments, and results are clearly written.

**Top Reasons To Reject The Paper:**

* Comparison with other benchmarks is mostly qualitative and could be stronger if it were more quantitative.
* The results can be expanded upon with some more details on the insights and significance of the comparisons.

---

### Official Review · Reviewer_ND6a · 2024-05-28
**The paper does bring up an important point of fair comparison and provides a useful tool, however it has limited novelty**

**Confidence:** 4
**Rating:** 4

**Detailed Feedback And Questions For Authors:**

## Summary

This paper proposes a convolution benchmarking tool which profiles end to end operation time including pre-convolution steps like analysis, data re-ordering, the in-convolution time and the post-convolution steps. The tool also provides a convset, which has convolutions sampled across many DL models.

## Feedback

While the tool seems to be useful to do a in-depth step by step analysis of convolution operation across algorithms, I think the novelty is limited to providing a conv-dataset and visualization of overheads. The roadmap for this tool seems to add data on cache hit/misses, memory accesses and TLB misses, which is nice but the tool feels too early and minimal now.

In the analysis of Sconv, the paper highlights that it finds a inefficiency in packing causing a slowdown for point-wise convolutions, however adding more labels, and metrics will aid in determining if this is a slowdown due to implementation or something fundamental resulting from the nature of the algorithm and the input shape. The tool has to provide such additional data to pinpoint the source of the slowdown to stand itself out from general profiling approaches.

Could you use a color palette with more contrast for Figure 3? It is hard to understand different components in the stacked bar graph. Also does the X axis represent a convolution ID in the dataset ?

**Top Reasons To Accept The Paper:**

- The paper argues that convolution benchmarking is ad-hoc and error-prone due to different layouts, pre and post processing overheads which vary based on the algorithm. Having a framework and a work set to measure true end to end time of the convolution operation is important.

- Convset is a useful dataset to assess general performance for a conv algorithm, and understand the relative efficiency over pointwise, regular and other convolutions.

**Top Reasons To Reject The Paper:**

- Limited novelty : The key novelty of the paper is limited to a C++ wrapper to measure and visualize different stages of convolutions across many shapes.  While it is a useful convenient tool, I do not see these features standing out when compared to traditional approaches.
  - The TM tool seems like a time measuring wrapper for different stages of convolution operation.

---

### Official Review · Reviewer_c8sC · 2024-05-29
**This paper presents ConvBench, a primitive-level benchmark for evaluating and comparing different convolution algorithms. Existing benchmarks frequently have limited scope and cannot provide a comprehensive comparison of convolution algorithms because of varying data layouts and system resources. ConvBench is proposed to address these issues by offering a comprehensive and fair comparison. The ConvBench benchmark tests 9243 convolution operations using 1097 real-world CNN models to address the limitations of existing benchmarks. A detailed graph of execution breakdown is provided to assess the performance of these operations. A performance insight and optimization opportunity are revealed in the study of Sliced Convolution (SConv).**

**Confidence:** 4
**Rating:** 6

**Detailed Feedback And Questions For Authors:**

1-The paper focuses on the SConv algorithm for evaluation.
2-The benchmark's performance evaluation is conducted on a specific hardware setup, which might not generalize to other systems. The paper does not provide results across different hardware/system setups.
3-The paper does not discuss the impact on the accuracy of the CNN models.
4-The paper does not discuss related comprehensively.
5-The paper may need to provide results for Memory and Power as they are becoming more important nowadays.
6-Authors are recommended to open the source code.
7-The results could be longer and more comprehensive.
8-The visualization and quality of the plots are low.
9-The paper also needs to talk extensively about the insights and lessons they learned from their experiment and results.

**Top Reasons To Accept The Paper:**

1-ConvBench provides comprehensive evaluations of convolution algorithms by evaluating a wide variety of convolution operations (9243) generated by a large number of CNN models (1097).
2-The convolution operations are derived from CNN models used in real-world situations, adding relevance and practicality to the benchmark.
3-By considering preprocessing and post-processing steps within convolution operations, ConvBench tries to find a fair comparison.

**Top Reasons To Reject The Paper:**

1-The benchmark's performance evaluation is conducted on a specific hardware setup, which might not generalize to other systems. The paper does not provide results across different hardware/system setups.
2- -The results could be longer and more comprehensive

---

### Official Review · Reviewer_EbG5 · 2024-05-30
**Review for ConvBench paper**

**Confidence:** 4
**Rating:** 4

**Detailed Feedback And Questions For Authors:**

Summary
The paper proposes a framework called "ConvBench" to evaluate various convolution implementation algorithms. The framework is designed to evaluate and compare 9243 2D convolution operations collected from 1097 real CNN models. It provides a brief explanation of the framework design, usage instructions, and related work, along with some experimental results. ConvBench primarily aims to help designers explore optimization opportunities in convolution algorithms. It includes pre-ordering and data preparation to provide fairer and more accurate analysis, which sets it apart from other state-of-the-art frameworks. However, the current version lacks hardware metrics and calculations. System-related metrics such as memory accesses and cache hit/miss rates are not considered but are planned for future inclusion. While the presentation of the paper is good, the proposed tool's value is limited due to the absence of vital hardware and system-related parameters.
Pros
	• Extensive Dataset: The collection of 9243 unique 2D convolution operations from 1097 CNN models is commendable. It provides a comprehensive dataset for evaluation.
	• Execution Breakdown and Performance Graphs: The framework’s ability to provide detailed execution breakdowns and performance graphs is insightful for identifying critical operations and optimization opportunities.
	• Consideration of Additional Operations: By taking into account operations such as strided/padding, data packing/unpacking, and data reordering, ConvBench distinguishes itself from other frameworks that ignore these important aspects.
Cons
	• Predominance of Pointwise Convolutions: A high percentage (60.8%) of the ConvSet is comprised of pointwise convolution operations, limiting the generalizability of the framework.
	• Lack of Cycle-Accurate Performance Results: The framework does not generate cycle-accurate performance results, and the experiments depend on specific evaluation setup mentioned in the section IV.
	• Unspecified Real-World Models: The paper mentions the use of seven common DL models but only specifies three (GoogleNet, Inception-V2, and Inception-V3), limiting transparency and comprehensiveness. Also very limited number of models compared to the dataset that has been already collected.
	• Focus on CNN Convolutions: The focus on convolutional layers within CNNs overlooks the growing importance of transformer architectures, which could also benefit from evaluation.
	• Omission of Hardware Accelerators and Sparsity Techniques: The framework does not consider techniques such as zero-skipping and sparsity-aware implementations or different arithmetic representations, which are crucial for accurate evaluation in modern hardware accelerators.
	• Isolated Convolution Evaluation: Evaluating convolution operations in isolation may not reflect real-world scenarios where other layers and data characteristics impact performance.

**Top Reasons To Accept The Paper:**

• Extensive Dataset: The collection of 9243 unique 2D convolution operations from 1097 CNN models is commendable. It provides a comprehensive dataset for evaluation.
	• Execution Breakdown and Performance Graphs: The framework’s ability to provide detailed execution breakdowns and performance graphs is insightful for identifying critical operations and optimization opportunities.
	• Consideration of Additional Operations: By taking into account operations such as strided/padding, data packing/unpacking, and data reordering, ConvBench distinguishes itself from other frameworks that ignore these important aspects.

**Top Reasons To Reject The Paper:**

• Predominance of Pointwise Convolutions: A high percentage (60.8%) of the ConvSet is comprised of pointwise convolution operations, limiting the generalizability of the framework.
	• Lack of Cycle-Accurate Performance Results: The framework does not generate cycle-accurate performance results, and the experiments depend on specific evaluation setup mentioned in the section IV.
	• Unspecified Real-World Models: The paper mentions the use of seven common DL models but only specifies three (GoogleNet, Inception-V2, and Inception-V3), limiting transparency and comprehensiveness. Also very limited number of models compared to the dataset that has been already collected.
	• Focus on CNN Convolutions: The focus on convolutional layers within CNNs overlooks the growing importance of transformer architectures, which could also benefit from evaluation.
	• Omission of Hardware Accelerators and Sparsity Techniques: The framework does not consider techniques such as zero-skipping and sparsity-aware implementations or different arithmetic representations, which are crucial for accurate evaluation in modern hardware accelerators.
Isolated Convolution Evaluation: Evaluating convolution operations in isolation may not reflect real-world scenarios where other layers and data characteristics impact performance.

---

### Decision · Program_Chairs · 2024-05-30

**Decision:**

Accept (Oral/Poster)

**Comment:**

Congratulations! We are pleased to inform you that your paper has been accepted for presentation at MLArchSys 2024. We look forward to your participation at the workshop. Further details regarding the schedule and format will be provided soon. See you at the workshop!